# Investigating the Development of STEM-Positive Identities of Refugee Teens in a Physics Out-of-School-Time Experience †

Tino Nyawelo [1,*], Sarah Braden [2], John N. Matthews [1], Jordan Gerton [1], Bolaji Bamidele [2], Melanie Valera Garcia [2], Raquel Goldrup [2], Ricardo Gonzalez [1] and Joseph Kiflom [1]

1   Department of Physics and Astronomy, University of Utah, Salt Lake City, UT 84112, USA; jnm@cosmic.utah.edu (J.N.M.); jordan.gerton@utah.edu (J.G.); ricardo.gonzalez@utah.edu (R.G.); u1053528@utah.edu (J.K.)
2   School of Teacher Education and Leadership, Utah State University, Logan, UT 84322, USA; sarah.braden@usu.edu (S.B.); a02384457@usu.edu (M.V.G.); rgoldrup8@gmail.com (R.G.)
*   Correspondence: tino.nyawelo@utah.edu
†   Presented at the 23rd International Workshop on Neutrinos from Accelerators, Salt Lake City, UT, USA, 30–31 July 2022.

**Abstract:** Refugee youth resettled in the United States experience two main barriers to long-term participation in STEM fields: (a) access to STEM skills and knowledge which is impacted by relocation and interrupted schooling, and (b) access to crafting positive learner identities in STEM as multilingual, multicultural, and multiracial youth. In this paper, we share a model for engaging refugee teens in cosmic ray research through constructing scintillator cosmic ray detectors, creating digital stories about cosmic rays, and hosting family and community science events where students share their learning with their families. This context serves as the site for ongoing ethnography exploring how refugee-background teens construct STEM-related identities and identifying supportive and unsupportive instructional practices. This paper summarizes the key program details and findings to date.

**Keywords:** STEM identity; informal science education; refugee; cosmic rays; particle physics

## 1. Introduction

Youth with refugee backgrounds account for approximately 6% of the U.S. K-12 student population [1]. For students labeled as refugees, socio-economic status, previous educational experience, and language learning stress can impact their academic success in the U.S [2]. Refugee student drop-out-of-school rates are twice as high as those of U.S.-born students [2,3]. Refugee youth may experience a gap in Science, Technology, Engineering, and Mathematics (STEM) skills and knowledge, and there can be conflict between the identities, roles, and responsibilities necessary for participation in their families and communities and those expected for success in STEM settings [4–6]. Because of these identity-based challenges, building skills and providing exposure to science learning *is necessary*, but alone is not *sufficient* for fostering refugee students' success in STEM disciplines.

In this paper, we examine the STEM-related identity work of multilingual refugee youth through a linguistic anthropological framework that examines identities in practice. We use the terms *identity work* and *identities in practice* to draw attention to the dynamic social and performative aspects of identity and to distinguish from approaches that theorize identity as being static and internal. Similarly, we use the term *STEM-related* to highlight the emic ways in which participants articulate their identities as opposed to *STEM identity*, which we view as an etic term. The following research questions guided the design of the program and continue to guide our ongoing research:

1.   How do students craft STEM-related identities across the various relationships, modes of interaction, and activities afforded by the program?

2.  How, when, and where do students perform discipline-specific identities versus more general STEM identities?

## 2. Theoretical Framework: Pathways of Identity Development

We theorize identity as being socially performed through conversation and thus instantiated in and through social interactions that unfold over pathways over time [7,8]. This approach allows the analyst to identify the semiotic resources, including linguistic practices, which typify identities of expertise in both local and broader societal contexts [9]. We explored how students in the program typified themselves and others as they learned physics, computing, and film making. We also investigated how students' social performances of their multifaceted identities (e.g., in terms of race, gender, etc.) are entangled with their processes of becoming in STEM. Within this linguistic ethnographic approach, metacommentary, or "metapragmatic discourse" [8], offers a window into how participants articulate and in so doing recreate social types or tropes as ideological cultural artifacts.

## 3. Methods

### 3.1. Context

We study students' STEM identity development by engaging youth in authentic cosmic ray research. In 1938, Pierre Auger and colleagues [10] were investigating atmospheric radiation using small particle detectors when they noticed that two detectors spaced some distance apart tended to detect particles at the same time. The particles seemed to be arriving in bunches spread out over large areas. Eventually, they realized that a much higher energy particle was colliding with a molecule in the atmosphere. This primary "cosmic ray" particle created an *air shower* (Figure 1) of secondary particles, whereby some of the particles reached the ground.

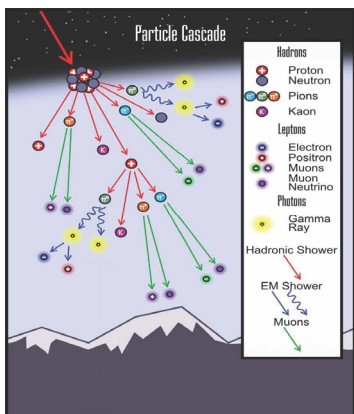

**Figure 1.** Schematic of air shower development.

To engage refugee youth in a meaningful physics research experience, we guided them through the process of building detectors which capture the activity of these secondary particles, and we mentored them through research projects in which they learned to analyze the data collected using this type of detector. Instructional design is informed by a culturally sustaining pedagogy [11] approach which focuses on identifying and leveraging students' cultural and linguistic assets for learning. Notably, due to the COVID-19 pandemic, the project began using Zoom with detector construction delayed. Over Zoom, the students learned about cosmic rays and coding through interactive presentations and they completed filming exercises to learn specific techniques. In order to create a space for the students to reflect on their experiences in cosmic ray research, we also engaged them in completing reflective activities and producing short films, or digital stories, which "are artistic and analytic demonstrations of how we come to know, name, and interpret cultural experience" [12], in this case, scientific research. The students experienced these activities through twice-weekly sessions during the school year (Years 1 and 2) and a 1-week

intensive summer experience. The program also hosts twice-yearly family and community events in which students share what they are learning with others. Specific activities for each of these three activity types are listed in Figures 2 and 3 contains pictures of students constructing detectors.

| Summary of Program Activities |
|---|
| **Cosmic Ray Research** <br> <u>Detector Construction</u>: Cleaning scintillator and light guide, preparing epoxy (mixing and air removal using vacuum), gluing the scintillator to the light guide, light-tight wrapping the scintillator and light guide, wiring for the PMT, installing the PMT to the light guide, and testing and calibrating the detectors. Students also tagged and decorated their detectors. <br> <u>Research Design and Data Analysis</u>: accessing data from the HiSPARC database, developing and refining a research question, identifying a dataset to answer a research question, graphing, interpreting data, and developing a scientific poster (this phase is still in progress). |
| **Digital Storytelling** <br> <u>Reflection Activities</u>: openly discussing stereotypes in STEM, reflecting on experiences learning science in school and in the afterschool program, constructing a self-portrait using labels, designing a personal logo, and writing an "I am" poem. <br> <u>Filming</u>: filming techniques, scripting, storyboarding, rehearsing, and editing. |
| **Family and Community Science Events** <br> Students share digital stories with parents and answer questions from the audience. Share pictures of students during detector building with parents and discuss students' work on the project, including a tour of completed detectors. |

**Figure 2.** Student activities.

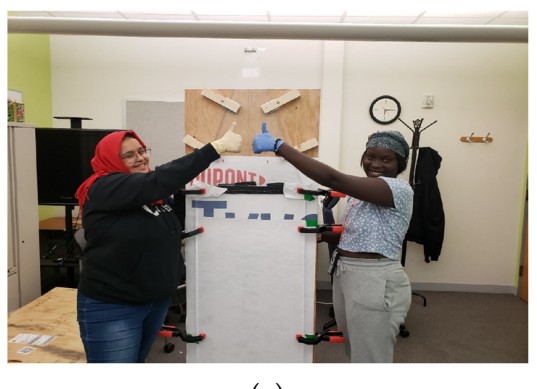

(**a**)

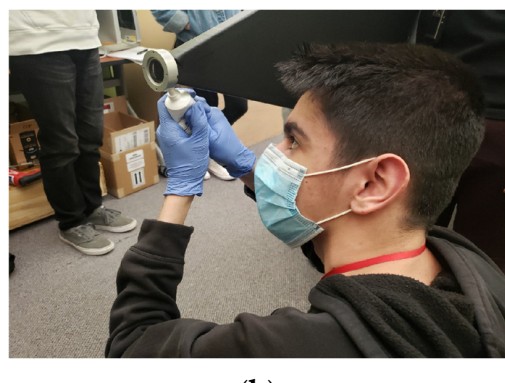

(**b**)

**Figure 3.** (**a**) Students celebrating the successful optical gluing of the light guide to the scintillator panel. (**b**) A student completing the light seal of a scintillator in preparation for attaching the photomultiplier tube.

### 3.2. Participants

Twenty-eight students (mostly 8th–12th grade) participated in the project in Year 1, with ten students continuing in Year 2 and thirteen new students joining the program. Students were from the following countries of origin: Nepal, Burma, Uganda, Sudan, South Sudan, Iraq, Ethiopia, Democratic Republic of the Congo, Bhutan, and Mexico.

### 3.3. Data Collection and Analysis

Video and audio recordings of student conversations, photos, field notes from participant observation, student artifacts, and semi-structured interviews were collected from spring 2021 to spring 2023. The emergent findings to date come from several componential analyses using ethnographic and discourse analytic methods as a means of chipping away at answering larger research questions. We applied a constant comparative [13] process which meant applying a recursive, reflexive process to identify patterns across data sources. This included both thematic content analysis and a linguistic anthropological approach to discourse analysis [8]. Applying these methods, we analyzed *who* students are becoming and *how* (through what linguistic practices) they construct a local persona within our program.

## 4. Findings and Implications

The preliminary analysis of students' conversations and interviews indicates that the program activities conducted through the strand of digital storytelling (which included reflective tasks) created a space for students to explicitly share and discuss aspects of their racial and linguistic identities, while also discussing their science and mathematics learning experiences. We believe that offering students these opportunities to grapple with and express their multifaceted identities in the context of a physics-focused afterschool program created a positive means for their racial and linguistic identities to become productively stitched with their STEM-related identities. By discussing their challenges with science learning in school, which included experiences of racism and decontextualized instruction prioritizing rote learning, the students found a place to receive validation and engage in more meaningful science education. In addition, for students who experienced success and joy in learning science in school and in other informal STEM learning environments, the reflective activities also gave them a space to discuss and receive validation for these experiences. Relatedly, the findings also suggest the importance of relationship building between the instructors and students to disrupt traditional power hierarchies and create a learning environment that supports students' agency in weaving youth culture into our program activities. We saw this in two primary areas: (a) student-generated digital stories which included playful stylistic choices reflecting an informal vlog style, and (b) support for the youth's engagement in play while constructing and decorating the cosmic ray detectors.

These findings have implications for other research teams aiming to engage "underrepresented" students in STEM learning. Namely, access to status quo science learning opportunities is not sufficient to engage linguistically, culturally, and racially diverse students whose needs are often not met in traditional schooling. To engage such learners in science learning, scientists and science instructors must be open to reimagining what might count as relevant disciplinary practices to include reshaping the social contexts of scientific research. This may include reshaping the power dynamics between scientists and students by engaging in age-appropriate playful practices directed by students, and relationship building.

**Author Contributions:** Conceptualization and methodology, T.N., S.B., J.G. and J.N.M.; formal analysis, data curation, S.B.; data collection and instruction, S.B., T.N., J.G., J.N.M., B.B., M.V.G., R.G. (Raquel Goldrup), J.K. and R.G. (Ricardo Gonzalez); writing—original draft preparation, T.N.; writing—review and editing, T.N. and S.B.; project administration and funding acquisition, T.N., S.B., J.G. and J.N.M. All authors have read and agreed to the published version of the manuscript.

**Funding:** This research is sponsored in part by the National Science Foundation Award 2005973.

**Institutional Review Board Statement:** Not applicable.

**Informed Consent Statement:** Not applicable.

**Data Availability Statement:** Not applicable.

**Acknowledgments:** We would like to thank Bob van Eijk, Jan-Willem van Holten, Kasper van Dam, Arne de Laat, and David Fokkema David, as well as the team at NIKHEF (National Institute for Subatomic Physics), for their valuable help in getting the project started and for their ongoing assistance. They also provided very valuable electronics, detectors, documentation, papers, and consultation for the project, as well as pleasant discussions. The assistance of Anita Orendt, Joe Breen, and the University of Utah's CHPC is greatly appreciated.

**Conflicts of Interest:** The authors declare no conflict of interest.

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
