# Peer review of "Investigating the Development of STEM-Positive Identities of Refugee Teens in a Physics Out-of-School-Time Experience†"

_psf, doi:10.3390/psf8010045_

Round 1

Reviewer 1 Report

In the manuscript 'Investigating the Development of STEM-Positive Identities of Refugee Teens in a Physics Out-of-School Time Experience' the authors describe an intervention in the STEM education of refugees from several countries. A course on cosmic rays defines the physical background. It consists of three parts, viz. the building of a detector, the reflective activities
(on persons working in STEM), a digital story telling via a film, and community
events. The research questions aim at identity development in STEM. The
manuscript provides a rough description of all parts of the intervention, and
the research method. Unfortunately no results are available so far.

Since the research questions could not be answered clearly so far, we do
not learn much about identity development in STEM at present. However,
the intervention and research method seem to be interesting and could
be an inspiration for other researchers. Thus, even though this is will
definitely not be one of the most high-ranking articles in Physical Sciences
Forum, I consider it in general to be interesting for publication in the
journal. The presented contents seems to be sound. However, I have a few
questions before I can provide a final recommendation.

It is a pity that we do not learn any answer on the research questions. If
the manuscript was worthwhile to be published, we should at least learn
how this can be extracted in the future. Unfortunately, the analysis section
does not tell much about how the study will be done. I think a few more
comments on this topic will be very helpful for the reader.

The last sentence concludes that 'analyses show the activities supported
"positive" STEM-related identity work for many students'. What are these
signs that allow for this conclusion? I think it would help the reader if
this is mentioned.

The data collection and analysis section mentions recordings of student
conversations, notes, and interviews as methods of collecting data. I think
it will be very interesting for the readers, which method was used in which
part of the intervention (detector building, storytelling, ...).

For me it is not completely clear how the physical background was used
throughout the whole intervention. Have the detectors been used? What was
the physical contents of the films? What did the students explain in the
community events? Was there any check of the effectiveness in learning the
physical contents?

It is mentioned that students were given creative freedom in the design of
the films. However they also did get help in all parts of the production.
How was the creative freedom ensured?

Minor comment:

The acronym INSPIRE is only used once in the first research question but never
defined in the manuscript. This should be included.

Author Response

Thank you, Reviewer 1, for your thoughtful comments. We’ve endeavored to improve the manuscript according to your feedback. Our specific responses detailing manuscript changes are attached.

Reviewer 2 Report

The manuscript proposes to explore the compelling sociological subject of refugee teens developing their identities within the context of STEM. Such a study holds considerable potential for creating a significant educational model with far-reaching benefits for similar students. The authors' two-year program, encompassing a range of activities and events involving the students, their families, and communities, is commendable.

Nonetheless, the primary issue lies in the research's seeming incompleteness, with scant results or conclusions presented for discussion. The Introduction and Methods sections are well articulated, yet the Data Collection & Analysis section is excessively succinct and unclear, causing reader uncertainty regarding the substantive exploration of identity formation via the data and materials amassed during the program.

Even though the authors state that "no single analysis to date fully answers the research questions", there is a conspicuous absence of any description of the analytic process undertaken. A positive atmosphere during events, although beneficial, does not constitute a definitive measure of the program's success in fostering STEM-positive identities among refugee teens.

Therefore, the manuscript leaves readers uncertain about the study's methodology and bereft of actionable insights or practical takeaways regarding the facilitation of STEM-positive identity formation among refugee teens.

Additional minor points for revision include:

1. Enhancing clarity by expanding the acronyms STEM and INSPIRE (Line 38) upon their initial mention.

2. In the Introduction, briefly define "STEM-related identity" and elaborate on its significance for student success in STEM disciplines.

3. Clarify the use of square brackets around "that" at Line 55.

4. Correct the extra space at Line 59.

5. Rephrase sentences at Line 95-96 and Line 99 for improved readability.

Author Response

We thank Reviewer 2 for their useful comments. We’ve thoroughly revised the manuscript according to the recommendations. Our responses to each reviewer comment are attached.

Round 2

Reviewer 1 Report

The authors have carefully considered the comments of the referees and revised the manuscript accordingly. In my view the revision is very extensive and very satisfactorily. I do not have further questions.

As mentioned in my last report I consider this article to be interesting for the readership of the Physical Sciences Forum. With the revisions included, I think that it has become even more interesting. In my view it can be published in the present form.

Reviewer 2 Report

The authors deserve commendation for their responsive efforts to address the feedback provided. There is a notable enhancement in the sections pertaining to methods, data analysis, and implications, thanks to the inclusion of more detailed descriptions and deeper discussion. Although the preliminary nature of the results still limits the paper's impact, the manuscript is sufficiently valuable to merit publication.

One minor revision I would suggest is that the current Figure 2 is actually a table rather than a figure.